# EgoExo-Gen: Ego-centric Video Prediction by Watching Exo-centric Videos

**Jilan Xu**[1,7], **Yifei Huang**[2,7], **Baoqi Pei**[3,7], **Junlin Hou**[4], **Qingqiu Li**[1], **Guo Chen**[5],
**Yuejie Zhang**[1,*], **Rui Feng**[1,*], **Weidi Xie**[6,7]

[1]School of Computer Science, Shanghai Key Lab of Intelligent Information Processing,
Shanghai Collaborative Innovation Center of Intelligent Visual Computing, Fudan University
[2]The University of Tokyo, [3]Zhejiang University, [4]Hong Kong University of Science and Technology,
[5]Nanjing University, [6]Shanghai Jiao Tong University, [7]Shanghai Artificial Intelligence Laboratory
`jilanxu18@fudan.edu.cn`

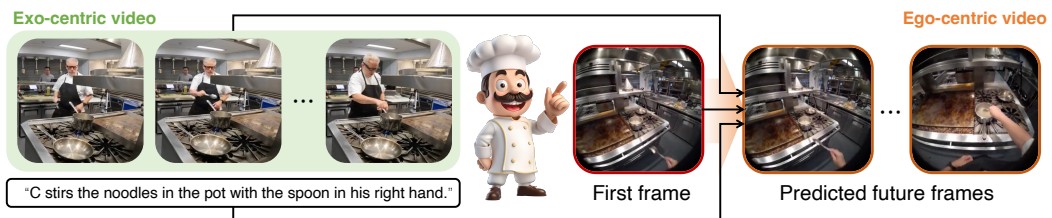

Figure 1: The cross-view video prediction task aims to predict future RGB frames of the ego-centric video, given the first ego-centric frame, a text instruction, and a synchronised exo-centric video.

## Abstract

Generating videos in the first-person perspective has broad application prospects in the field of augmented reality and embodied intelligence. In this work, we explore the cross-view video prediction task, where given an exo-centric video, the first frame of the corresponding ego-centric video, and textual instructions, the goal is to generate future frames of the ego-centric video. Inspired by the notion that hand-object interactions (HOI) in ego-centric videos represent the primary intentions and actions of the current actor, we present EgoExo-Gen that explicitly models the hand-object dynamics for cross-view video prediction. EgoExo-Gen consists of two stages. First, we design a cross-view HOI mask prediction model that anticipates the HOI masks in future ego-frames by modeling the spatio-temporal ego-exo correspondence. Next, we employ a video diffusion model to predict future ego-frames using the first ego-frame and textual instructions, while incorporating the HOI masks as structural guidance to enhance prediction quality. To facilitate training, we develop an automated pipeline to generate pseudo HOI masks for both ego- and exo-videos by exploiting vision foundation models. Extensive experiments demonstrate that our proposed EgoExo-Gen achieves better prediction performance compared to previous video prediction models on the Ego-Exo4D and H2O benchmark datasets, with the HOI masks significantly improving the generation of hands and interactive objects in the ego-centric videos.

## 1 Introduction

This paper considers the task of animating an ego-centric frame based on a third-person (exo-centric) video captured simultaneously in the same environment. As illustrated in Fig. 1, given the first frame of an ego-centric video, a textual instruction, and a synchronised exo-centric video (Grauman et al., 2024; Kwon et al., 2021; Jia et al., 2020; Sener et al., 2022), the goal is to predict the subsequent frames in ego-view. Exo-centric views typically provide broader environmental context and body kinetics (Carreira & Zisserman, 2017; Miech et al., 2019), but are less focused on fine-grained

---

* Corresponding author.

actions. In contrast, ego-centric views center on the hands of the camera wearer and the interacting objects (Grauman et al., 2022; Damen et al., 2018), which are critical for tasks like manipulation and navigation (Nair et al., 2022; Bharadhwaj et al., 2023; Rhinehart & Kitani, 2017; Shah et al., 2022). The challenge in cross-view video prediction arises from the significant perspective shift between these views, as the future frames must align with both the contextual environment of the first frame and the actor's motion as indicated by the textual instruction and the exo-centric video.

Bridging these two perspectives offers two key benefits: *first*, it allows agents to build a robust ego-centric world model from third-person demonstrations, enabling them to translate broader scene information into a first-person perspective (Huang et al., 2024a; Luo et al., 2024b); *second*, it enables the agents to think from the humans' perspective by aligning their viewpoint with human users, improving their ability to anticipate future actions and make more informed decisions. This capability is particularly valuable in real-time applications such as augmented reality (AR) (Huang et al., 2018; 2024b) and robotics (Wang et al., 2023a), where agents must synchronize their actions with humans or the environment. By learning to translate exo-centric video into accurate ego-centric frames, agents can become more adaptive when engaging in dynamic, real-world environments.

Recent researches in video prediction models have shown tremendous progress (Xing et al., 2023; Chen et al., 2023; Ren et al., 2024; Guo et al., 2023a). These models mainly rely on the first frame and textual instructions as inputs for diffusion models (Rombach et al., 2022b; Song et al., 2020), with a primary focus on generating videos from an exo-centric perspective. Regarding ego-centric video prediction, (Gu et al., 2023; Xing et al., 2024) have explored the decomposition of text instructions, but they lack customised design for the ego-centric view. In the context of cross-view video generation, specialised models have been developed (Luo et al., 2024a;b). For example, (Luo et al., 2024a) leveraged head trajectory data from ego-centric videos to generate optical flow and occlusion maps in exo-centric videos, while (Luo et al., 2024b) estimated hand poses in ego-centric views to guide conditional ego-video generation. However, these approaches rely on 3D scene reconstruction (Schonberger & Frahm, 2016; Tschernezki et al., 2024) or precise human annotations (Kwon et al., 2021), limiting their scalability and generalization ability across diverse scenarios.

In this paper, we propose **EgoExo-Gen**, a cross-view video prediction model that generates future ego-centric video frames by explicitly modeling hand-object dynamics. EgoExo-Gen employs a two-stage approach: *first*, it predicts semantic hand-object interaction (HOI) masks in ego-centric view, and *second*, it uses these masks as condition, driving a diffusion model to produce the corresponding RGB frames. For the cross-view HOI mask prediction model, it takes an exo-centric video and the first frame of an ego-centric video as inputs, and predicts HOI masks for future ego-centric frames. To handle the drastic change in viewpoints, we design an ego-exo memory attention module that captures spatio-temporal correspondences between the two views, enabling the model to infer HOI masks even when the current ego-centric frame is not visible. The predicted HOI masks are then integrated into the diffusion model, guiding the generation of accurate future ego-centric frames. To ensure scalable training and minimise reliance on manual annotations, we also develop a fully automated HOI mask annotation pipeline for both ego-centric and exo-centric videos by leveraging powerful vision foundation models (Khirodkar et al., 2024; Kirillov et al., 2023; Ravi et al., 2024). This combination of methods allows EgoExo-Gen to generate high-quality ego-centric videos while significantly improving scalability and generalisation across diverse environments.

We conduct extensive experiments on the cross-view video benchmark datasets, *i.e.*, Ego-Exo4D (Grauman et al., 2024) and H2O (Kwon et al., 2021) that include rich and diverse hand-object interactions and shooting environment. Experimental results show that EgoExo-Gen significantly outperforms prior video prediction models (Chen et al., 2023; Ren et al., 2024; Gu et al., 2023) and improves the quality of predicted videos by leveraging hand and object dynamics. Also, EgoExo-Gen demonstrates strong zero-shot transfer ability on unseen actions and environments.

## 2 METHODOLOGY

### 2.1 PROBLEM FORMULATION

This paper considers a challenging problem of animating one ego-centric frame, based on an exo-centric video, captured at the same time and in the same environment. Specifically, given the exo-centric video with $N$ frames, $\mathcal{V}_{\text{exo}} = \{x_1, \ldots, x_N\}$, the first ego-centric frame $g_1$, and a text instruc-

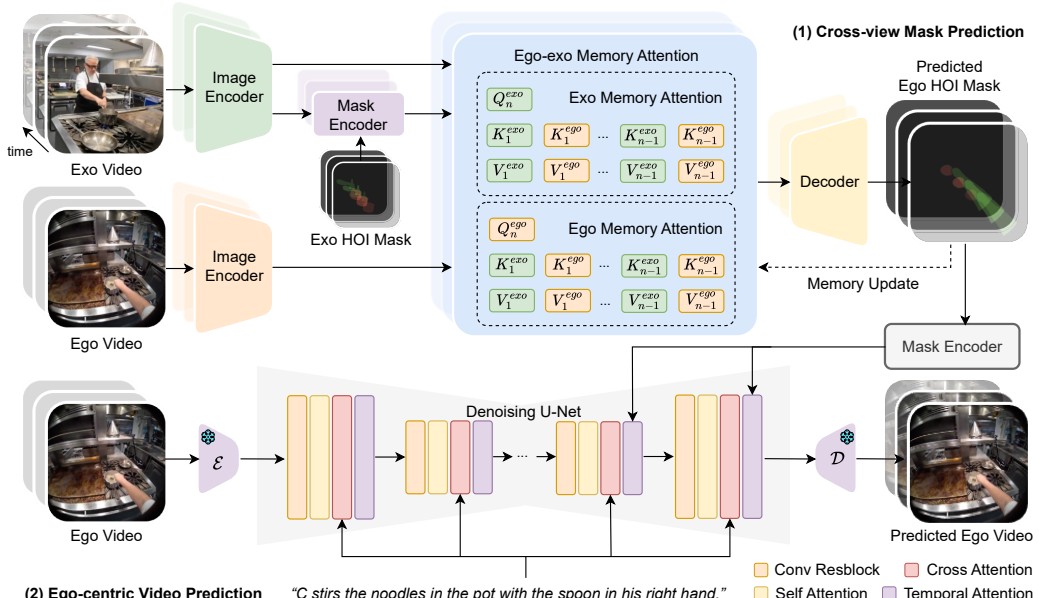

Figure 2: **An overview of EgoExo-Gen.** Given an exo-centric video, a text instruction, and the first frame of an ego-centric video, (1) a cross-view mask prediction model first anticipates the hand-object masks of the unobserved future frames; (2) an HOI-aware video diffusion model then predicts future frames of an ego-centric video by incorporating the predicted hand-object masks.

tion $\mathcal{T}$, our goal is to predict the corresponding ego-centric video, $\mathcal{V}_{\text{ego}} = \{g_1, \ldots, g_N\}$. Notably, the camera poses and depth information for both ego-centric and exo-centric videos are not available, encouraging the development of methods that can generalise across different cameras and locations. *Note that*, unlike the cross-view correspondence benchmark (Grauman et al., 2024) and exo-to-ego image translation (Luo et al., 2024b), our considered problem only assumes the availability of one starting frame in ego-centric view at test time, which makes this task significantly more challenging.

**Overview.** We propose EgoExo-Gen, that enables to animate one ego-centric video frame based on its corresponding exo-centric video, by explicitly modeling the hand-object dynamics. As illustrated in Fig. 2, our proposed model consists of: (1) a cross-view mask prediction network $\Phi_{\text{seg}}$ that estimates the spatio-temporal hand-object segmentation masks of the ego-video ($\hat{\mathcal{M}}_{\text{ego}}$); (2) an HOI-aware video diffusion model $\Phi_{\text{gen}}$ for predicting the RGB ego-frames ($\hat{\mathcal{V}}_{\text{ego}}$), given the hand-object masks; (3) a fully automated pipeline to generate ego-exo HOI segmentation masks for training both models. At inference time, the overall process can be formulated as:

$$\hat{\mathcal{M}}_{\text{ego}} = \Phi_{\text{seg}}(\mathcal{V}_{\text{exo}}, g_1), \quad \hat{\mathcal{V}}_{\text{ego}} = \{\hat{g}_2, \ldots, \hat{g}_N\} = \Phi_{\text{gen}}(g_1, \mathcal{T}, \hat{\mathcal{M}}_{\text{ego}}). \quad (1)$$

## 2.2 CROSS-VIEW MASK PREDICTION

This section details the cross-view mask prediction model, which translates the hand-object masks from an exo-centric video into ego-centric view. Specifically, the model takes as input: (i) exo-centric video, $\mathcal{V}_{\text{exo}}$; (ii) frame-wise hand object masks of the exo-centric videos, $\mathcal{M}_{\text{exo}} \in \{0, 1, 2\}^{N \times H \times W}$, including background, hand, and object; and (iii) the first frame of an ego-centric video, $g_1$. The desired outputs are the corresponding frame-wise hand-object masks in ego-centric view, $\hat{\mathcal{M}}_{\text{ego}}$, generated in an auto-regressive manner. In the following section, we only present the overall pipeline at training time, while during *inference*, we only assume the availability of one starting frame in the ego-centric view, thus replacing the ego-centric video with the first ego-centric frame, $\mathcal{V}_{\text{ego}} \rightarrow g_1$, and all zero images for subsequent ego-centric frames.

**Image and mask feature encoding.** For the image encoder, we adopt a shared ResNet (He et al., 2016) with the last convolution block removed. The encoder takes as input the exo- and ego-frame, and outputs the exo- and ego-features respectively, $\bar{x}_n, \bar{g}_n = \Phi_{\text{enc}}(x_n), \Phi_{\text{enc}}(g_n)$. The mask encoder takes the concatenation of the current exo-frame and exo-mask as input and encodes them with

another ResNet. Then it fuses with the exo-image feature via a CBAM block (Woo et al., 2018), and produces the exo-mask feature, $\overline{m}_n^x = \text{CBAM}(\overline{x}_n, \Phi_{\text{mask}}(x_n, m_n^x))$.

**Ego-exo memory attention.** As shown in Fig. 2, the core of this module is a memory bank, that enables to predict the HOI mask features for ego-centric videos from corresponding exo-centric frames. The model operates in an auto-regressive manner, for example, when generating for the $n^{th}$ ego-frame, the corresponding frame feature from exo-centric video is treated as `query`, the historical exo- and ego-frame features as `keys`, *e.g.*, $\overline{x}_1, \overline{g}_1, \ldots, \overline{x}_{n-1}, \overline{g}_{n-1}$, and corresponding mask features as `values`, *e.g.*, $\overline{m}_1^x, \hat{\overline{m}}_1^g, \ldots, \overline{m}_{n-1}^x, \hat{\overline{m}}_{n-1}^g$:

$$\mathcal{Q}^{\text{exo}} = W_q^{\text{T}} \overline{x}_n, \ \mathcal{K} = W_k^{\text{T}}[\overline{x}_1, \overline{g}_1, \ldots, \overline{x}_{n-1}, \overline{g}_{n-1}], \ \mathcal{V} = W_v^{\text{T}}[\overline{m}_1^x, \hat{\overline{m}}_1^g, \ldots, \overline{m}_{n-1}^x, \hat{\overline{m}}_{n-1}^g], \quad (2)$$

where $\hat{\overline{m}}_1^g, \ldots, \hat{\overline{m}}_{n-1}^g$ refer to mask features of earlier predictions; $W_q, W_k, W_v$ are linear transformations and [] denotes concatenation along an additional time dimension. The mask feature for the $n^{th}$ ego-frame can thus be obtained with cross-attention:

$$\mathcal{Z} = W_{\text{out}}^{\text{T}}(\mathcal{AV}), \quad \text{where } \mathcal{A} = \text{softmax}(\mathcal{Q}^{\text{exo}}\mathcal{K}^{\text{T}}/\sqrt{D_3}),$$

and the softmax operation is performed over the time dimension. At training time, as all frames in an ego-centric video are visible, we also obtain a residual memory output $\mathcal{Z}'$ by leveraging ego-visual features as query, *i.e.*, $\mathcal{Q}^{\text{ego}} = W_q^{\text{T}} \overline{g}_n$. The final output of the memory attention module is $\mathcal{Z}'' = \alpha \mathcal{Z}' + (1-\alpha)\mathcal{Z}$, where $\alpha$ anneals from 1.0 to 0.0 at training stage. We observe that such strategy helps the model training in the early stage, and eventually learns to predict the mask features for all ego-centric frames at inference time.

**Mask decoder.** The mask decoder consists of a stack of upsampling blocks, each consisting of two convolution layers followed by a bilinear upsampling operation (Oh et al., 2019; Cheng & Schwing, 2022). It accepts the output from the ego-exo memory attention module $\mathcal{Z}''$, and fuse it with multi-scale ego-features at each block, similar to UNet (Ronneberger et al., 2015) (not shown in Fig. 2). The decoding process can be simplified as: $\hat{m}_n^g = \Phi_{\text{dec}}(\mathcal{Z}'', \overline{g}_n^4, \overline{g}_n^8, \overline{g}_n^{16})$, where $\overline{g}_n^j \in \mathbb{R}^{H/j \times W/j \times d_j}$ refers to multi-scale feature from the image encoder with a downscale of $j$.

**Memory store and update.** As aforementioned, our ego-exo memory bank stores the key and value information of the past spatial image and mask features, respectively. At each time $n$, we encode the predicted ego-mask into ego-mask feature, $\hat{\overline{m}}_n^g = \text{CBAM}(\overline{g}_n, \Phi_{\text{mask}}([g_n, \hat{m}_n^g]))$, and store it into the ego/exo memory bank along with image and mask features, *i.e.*, $\overline{x}_n, \overline{g}_n$ and $\overline{m}_n^x$. At test time, we consolidate the memory and discard obsolete features while always retaining the reliable features of the first frame following (Cheng & Schwing, 2022; Ravi et al., 2024), but we do not emphasise this process as a contribution of our approach.

## 2.3 HOI-AWARE VIDEO DIFFUSION MODEL

The objective of the HOI-aware video diffusion model is to generate the subsequent ego-video frames $\{\hat{g}_2, \ldots, \hat{g}_N\}$, given the first frame $g_1$, text instruction $\mathcal{T}$ and the predicted ego-centric hand-object masks $\hat{\mathcal{M}}_{\text{ego}}$.

**Base architecture.** Following (Rombach et al., 2022b), the diffusion and denoising processes in are conducted in the latent space. $\Phi_{\text{gen}}$ consists of (i) a pre-trained VAE encoder $\mathcal{E}$ and decoder $\mathcal{D}$ for per-frame encoding/decoding of the ego-centric video; (ii) a mask encoder $\psi_{\text{mask}}$ for encoding hand-object masks; (iii) a denoising UNet $\epsilon_\theta$, parameterised by $\theta$. Following (Chen et al., 2023; Ren et al., 2024), the UNet architecture is constructed based on pre-trained text-to-image models (Rombach et al., 2022a). As illustrated in Fig. 2, in each down/upsampling block, a ResNet block, spatial self-attention and cross-attention layers are stacked. An additional temporal attention layer is appended to model the cross-frame relationship.

**First frame and text guidance.** The first frame $g_1$ of the ego-video is injected into the diffusion model to enable visual context guidance. Formally, the model takes as input the concatenation of three items ($\overline{z}$): (1) the corrupted noisy video frames; (2) the un-corrupted VAE feature representation of the video, with features of $2^{nd}$ to $N^{th}$ frames set to zero; and (3) a temporal mask $[1, 0, \ldots 0]^N$ indicating the visibility of the first frame. The text instruction $\mathcal{T}$ is tokenised and encoded via a CLIP text encoder (Radford et al., 2021), which is then injected to the UNet via cross-attention.

**HOI mask guidance.** To incorporate hand-object masks obtained from the cross-view mask prediction model into video diffusion model, $\{\hat{m}_1^g, \ldots, \hat{m}_N^g\}$, we encode them via a separate lightweight mask encoder ($\psi_{\text{mask}}$) and insert the features into the denoising UNet. $\psi_{\text{mask}}$ contains several downsampling blocks, each consists of a ResNet block followed by a temporal attention layer. It takes as input the HOI masks and outputs the multi-scale spatio-temporal feature maps:

$$h^4, h^8, h^{16}, h^{32} = \psi_{\text{mask}}([\hat{m}_1^g, \ldots, \hat{m}_N^g]), \tag{3}$$

where the mask feature $h^j \in \mathbb{R}^{H/j \times W/j \times C_j}$. In each decoder block, the mask features are fused with the latent features via element-wise addition, and are then fed to the temporal attention layer after linear projection. The training objective of the HOI-aware video diffusion model is denoted as:

$$\mathcal{L}_{\text{hoi\_diffusion}} = \mathbb{E}_{t, g \sim p_{\text{data}}, \epsilon \sim \mathcal{N}(0, \mathbf{I})} ||\epsilon - \epsilon_\theta(\bar{z}, \mathcal{T}, h, t)||_2^2, \tag{4}$$

where $t$ refers to the diffusion timestamp and $\epsilon \sim \mathcal{N}(0, \mathbf{I})$ denotes the Gaussian noise.

### 2.4 DATA, TRAINING AND INFERENCE PIPELINE

This section first describes an automatic procedure to generate ego-centric and exo-centric hand-object masks, that enables to move beyond reliance on existing labeled datasets and extend to paired ego and exo datasets in the wild. Then, we outline the training and inference processes.

**Ego-Exo HOI mask construction.** Our annotation pipeline is depicted in Fig. 3. For ego-centric videos, we first employ EgoHOS (Zhang et al., 2022) for per-frame segmentation of hands and interactive objects. Despite good results for individual frames, EgoHOS fails to capture the temporal dynamics between frames. To address this, we leverage the initial hand-object masks generated by EgoHOS as prompts for SAM-2 (Ravi et al., 2024), enabling object tracking across frames and ensuring mask consistency throughout the video. For exo-centric videos, EgoHOS proves inadequate for segmentation. Instead, we use 100DOH (Shan et al., 2020) for per-frame detection of hands and interactive objects. Additionally, we employ the human foundation model Sapiens (Khirodkar et al., 2024) to segment key regions, *i.e.*, left and right hands, forearms, and upper arms. For ego-centric videos, the bounding boxes and masks serve as prompts for SAM-2 to ensure consistent cross-frame tracking.

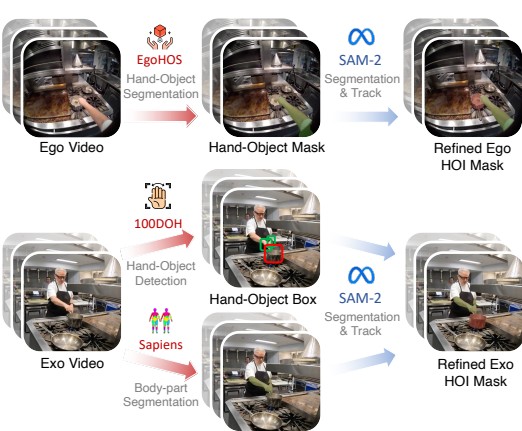

Figure 3: **Ego-Exo mask annotation pipeline.** We first perform frame-wise annotation with hand-object detector/segmentor, and prompt SAM-2 to track HOI masks in the video.

**Training stage.** To train the cross-view mask prediction model, we employ the sum of binary cross-entropy loss and dice loss. As the training progresses, we gradually mask ego-frames to encourage the model to learn cross-view correspondence when ego-frames are unobserved. The training process for the diffusion model is divided into two phases. In the first phase, we train the UNet backbone on ego-centric videos, using the first frame and textual instructions as conditions, without incorporating hand-object interactions (HOI). In the second phase, we freeze the UNet backbone and train the mask encoder and the linear projection layer, guided by Eq. 4. This phase utilises ego hand-object masks generated by our automated data pipeline, ensuring high-quality mask predictions.

**Inference stage.** The cross-view mask prediction model takes the exo-centric video, exo-centric HOI masks, and the first frame of the ego-centric video as inputs, and predicts ego-centric HOI masks for all subsequent frames. These predicted ego-centric HOI masks, along with the first frame of the ego-centric video, are then fed into the diffusion model to generate future ego-centric frames.

## 3 EXPERIMENTS

**Dataset.** We choose the Ego-Exo4D dataset (Grauman et al., 2024) in our experiment. Ego-Exo4D is the largest multiview dataset with 1,286 hours of video recorded worldwide. Each ego-centric

Table 1: Comparison of video prediction models on Ego-Exo4D and zero-shot transfer to H2O.

| Method | Ego-Exo4D | | | | Ego-Exo4D→H2O | | | |
|---|---|---|---|---|---|---|---|---|
| | SSIM ↑ | PSNR ↑ | LPIPS ↓ | FVD ↓ | SSIM↑ | PSNR↑ | LPIPS↓ | FVD↓ |
| SVD (Blattmann et al., 2023a) | 0.459 | 16.481 | 0.346 | 1120.680 | 0.394 | 14.310 | 0.338 | 1871.551 |
| Seer (Gu et al., 2023) | 0.376 | 15.468 | 0.527 | 1593.701 | 0.430 | 15.739 | 0.418 | 2272.068 |
| DynamiCrafter (Xing et al., 2023) | 0.457 | 15.880 | 0.434 | 1233.479 | 0.484 | 16.340 | 0.251 | 1657.931 |
| SparseCtrl (Guo et al., 2023a) | 0.474 | 16.455 | 0.355 | 1239.495 | 0.360 | 11.495 | 0.522 | 2566.101 |
| SEINE (Chen et al., 2023) | 0.518 | 17.680 | 0.321 | 1063.458 | 0.421 | 14.944 | 0.339 | 1797.965 |
| ConsistI2V (Ren et al., 2024) | 0.532 | 18.318 | 0.351 | 1109.314 | 0.458 | 16.188 | 0.276 | 1740.031 |
| EgoExo-Gen | **0.537** | **18.395** | **0.311** | **1031.693** | **0.486** | **16.424** | **0.240** | **1360.477** |

video has at least four exo-centric videos taken simultaneously in the same environment. We select videos belonging to the cooking scenario (564 hours taken under 60 different locations) as these videos contain rich hand-object interactions. The training set contains 33,448 video clips with an average duration of 1 second. Each video clip is paired with a narration (*e.g.*, *C drops the knife on the chopping board with his right hand.*) with start and end timestamps. Notably, the Ego-Exo4D cross-view relation benchmark contains paired ego-exo masks of specific objects, however, these objects are not guaranteed to be interacting objects, and the hand masks are not annotated. In contrast, our automatic annotation process includes both hands and interactive objects, offering greater scalability. We sample 1,000 video clips from the validation set, from which we select 500 video clips and annotate them with HOI masks to evaluate the performance of the mask prediction model. The training and validation sets have distinct takes, posing challenges to the model's generalisation ability on unseen subjects and locations. To evaluate the model's zero-shot transfer ability, we also adopt H2O (Kwon et al., 2021), an ego-exo HOI dataset focusing on tabletop activities (*e.g.*, squeeze lotion, grab spray). The validation set of H2O is composed of 122 clips with action labels.

**Implementation details.** We train our cross-view mask prediction model for 30 epochs with a batch size of 32 using the Adam optimizer. The initial learning rate is set to of $10^{-5}$. We sample 16 frames with a fixed spatial resolution of 480×480 for both ego-centric and exo-centric videos. We select one (out of four) exo-centric video with the highest proportion of exo-hand masks as the optimal viewpoint, effectively minimising occlusion issues. For the video diffusion model, we train both stages for 10 epochs with a batch size of 32 and a fixed learning rate of $10^{-4}$. We initialise our model with SEINE (Chen et al., 2023) pre-trained on web-scale video-text pairs, and train the model with 16 sampled frames with resolution 256×256. During inference, we adopt the DDIM sampler (Song et al., 2020) with 100 steps in our experiments.

**Evaluation metrics.** Following (Grauman et al., 2024), we evaluate the cross-view mask prediction model over three metrics: Intersection over Union (IoU), Contour Accuracy (CA), and Location Error (LE). IoU assesses the overlap between the Ground-truth mask and prediction; CA measures the shape similarity of the masks after applying translation to align their centroids; LE represents the normalised distance between the centroids of the predicted and ground-truth masks. Regarding the evaluation of video prediction models, we adopt SSIM (Wang et al., 2004), PSNR (Hore & Ziou, 2010), LPIPS (Zhang et al., 2018), and FVD (Unterthiner et al., 2018).

### 3.1 QUANTITATIVE COMPARISON

**Performance on Ego-Exo4D.** We compare our method with prior video prediction models with (i) first frame as condition (SVD (Blattmann et al., 2023a)); (ii) both text and first frame as conditions, including Seer (Gu et al., 2023), DynamiCrafter (Xing et al., 2023), SparseCtrl (Guo et al., 2023a), SEINE (Chen et al., 2023) and ConsistI2V (Ren et al., 2024). We finetune all these models on Ego-Exo4D except for SVD, which we found its zero-shot performance is better. As shown in Table 1, the fine-tuned ConsistI2V (Ren et al., 2024) and SEINE (Chen et al., 2023) models achieve comparably higher accuracy over the video prediction models, especially on SSIM and LPIPS, respectively. EgoExo-Gen consistently outperforms prior methods on all metrics, highlighting the benefits of explicit modeling the hand-object dynamics in video prediction models.

**Zero-shot transfer to H2O.** We evaluate our model's generalisation ability on unseen data distribution, *i.e.*, H2O (Kwon et al., 2021), containing synchronised ego-centric and exo-centric videos of hand-object interactions performing tabletop activities. The distinction of shooting location, context environment, object, and action categories compared to Ego-Exo4D poses challenges to model transfer. As revealed in Table 1 right, most methods achieve a relative high FVD (*i.e.*, low perfor-

mance). Our approach surpasses these models by effectively modeling the movement of hands and interactive objects, even when directly transferring from Ego-Exo4D to H2O without re-training.

Table 2: Comparison on different hand-object conditions.

| Method | SSIM↑ | PSNR↑ | FVD↓ |
|---|---|---|---|
| No mask | 0.518 | 17.681 | 1063.458 |
| Random | 0.528 | 18.132 | 1089.458 |
| Object only | 0.548 | 18.707 | 883.872 |
| Hand only | 0.565 | 19.065 | 851.705 |
| Hand object (ours) | **0.571** | **19.212** | **836.033** |
| Left right hand object | 0.569 | 19.188 | 839.349 |

Table 3: Comparison on different modalities as condition for video prediction.

| ID | Frame | Text | HOI | SSIM↑ | PSNR↑ | FVD↓ |
|---|---|---|---|---|---|---|
| 1 | ✓ | | | 0.477 | 16.628 | 1205.598 |
| 2 | | ✓ | | 0.229 | 8.579 | 3133.769 |
| 3 | ✓ | ✓ | | 0.518 | 17.680 | 1063.458 |
| 4 | ✓ | | ✓ | 0.555 | 18.930 | 864.739 |
| 5 | ✓ | ✓ | ✓ | **0.571** | **19.212** | **836.033** |

## 3.2 ABLATION STUDIES AND DISCUSSIONS

**Analysis on the HOI condition.** We compare different hand-object conditions in Table 2. To preserve the mask quality, here we apply the hand-object masks extracted from the future video frames instead of cross-view mask predictions. As shown in the first row, a baseline video prediction model without mask conditions only achieves 0.518 SSIM and 17.681 PSNR. Random mask does not make improvement over the baseline, with higher SSIM/PSNR but lower LPIPS/FVD. Applying either hand-only or object-only masks yield significant improvement over the baseline model. Our default choice of hand-object masks achieves the best results on all metrics, while distinguishing between left and right hands does not yield further performance gain. These results indicate that the fine-grained control of both hands and interacting objects is crucial for ego-centric video prediction.

**Analysis on conditioning modalities.** Table 3 lists the comparison results of applying different modalities as condition for the video prediction task. We again apply the HOI masks extracted from the future video to guarantee the mask quality. By default, the model takes as input the first frame and text instruction as conditions and predicts the subsequent frames (ID-3). Discarding text input (ID-1) makes the predicted video no longer conforms to human instructions, leading to less controllability. Similarly, the removal of the first frame as a control condition (ID-2) turns the model into a Text-to-Video (T2V) model. In this case, the model fails to generate actions in current scene context, and thus the performance degrades significantly (ID-1 vs. ID-3). When combining first frame and HOI mask, the model achieves superior performance than the baseline (ID-3 vs. ID-4). This indicates the structural control of the generation appears more effective than text instructions, as the hand-object movement can serve as a valuable cue for inferring the current action semantics. On the opposite, given a textual instruction, there can be multiple ways in which the hands and objects may actually interact and move. ID-5 shows that combining all modalities as conditions yields the best prediction performance.

**Analysis on the cross-view mask prediction.** We investigate the impact of ego memory attention (ego feature as query) and exo memory attention (exo feature as query) on cross-view mask prediction and video prediction in Table 4. The 1st row shows an oracle performance, where the future frames are visible to the cross-view mask prediction model. High accuracy on both segmentation and generation tasks can be observed as expected. As listed in the 2nd row, the model using ego-memory attention only yields low segmentation results. Despite its good segmentation on the visible first frame, it struggles with subsequent frames, as it can only use zero-image features as queries due to the invisibility, making it difficult to effectively aggregate historical information. In contrast, when using only exo-memory attention, the model can make better mask predictions than ego-memory attention on unobserved frames, resulting in overall better performance. Combining ego- and exo-memory attention (detailed in Sec. 2.2) assists the model training at early stages, with an enhanced prediction ability for segmenting hands and objects in both observed and unobserved frames, and subsequently improves the video prediction model.

**Analysis on exocentric video clues.** EgoExo-Gen decouples the cross-view video prediction task into cross-view learning (via a cross-view mask prediction model) and video prediction (via an HOI-aware video diffusion model). We modify our model to enable the incorporation of exocentric information in a single model, by replacing the ego-centric hand-object mask condition with either the original exo-centric RGB frames or the exo-centric hand-object masks. As observed in Table 5, both approaches obtain sub-optimal performance, indicating the difficulty of learning exo-ego trans-

Table 4: Comparison of the Ego/Exo memory attention (Ego-MA/Exo-MA) on cross-view mask prediction and video prediction.

| Method | Exo-MA | Ego-MA | Segmentation | | | Generation | | | |
|---|---|---|---|---|---|---|---|---|---|
| | | | IoU% ↑ | CA ↑ | LE ↓ | SSIM ↑ | PSNR ↑ | LPIPS ↓ | FVD ↓ |
| *w/ future* | | | | | | | | | |
| EgoExo-Gen | - | - | 63.594 | 0.715 | 0.037 | 0.561 | 18.909 | 0.284 | 859.444 |
| *w/o future* | | | | | | | | | |
| EgoExo-Gen | ✗ | ✓ | 4.630 | 0.048 | 0.182 | 0.529 | 18.143 | 0.322 | 1096.771 |
| EgoExo-Gen | ✓ | ✗ | 14.655 | 0.188 | 0.086 | 0.534 | 18.241 | **0.308** | 1049.182 |
| EgoExo-Gen (ours) | ✓ | ✓ | **20.315** | **0.207** | **0.082** | **0.537** | **18.395** | 0.311 | **1031.693** |

lation and video prediction in a single video diffusion model. In contrast, hand-object movement in the ego-centric view provides explicit pixel-aligned visual clues to improve the video prediction.

Table 5: Comparison on the incorporation of exo-centric information.

| Method | SSIM↑ | PSNR↑ | FVD↓ |
|---|---|---|---|
| *w/o exo* | | | |
| Baseline | 0.518 | 17.680 | 1063.548 |
| *w/ exo* | | | |
| Exo RGB | 0.525 | 17.901 | 1094.316 |
| Exo HOI | 0.529 | 18.013 | 1090.517 |
| Ego HOI (Ours) | **0.537** | **18.395** | **1031.693** |

Table 6: Comparison on the HOI mask quality in the annotation pipeline.

| Method | SSIM↑ | PSNR↑ | FVD↓ |
|---|---|---|---|
| *Hand only* | | | |
| EgoHOS (Zhang et al., 2022) | 0.562 | 19.001 | 865.407 |
| EgoHOS+SAM-2 | **0.565** | **19.065** | **851.705** |
| *Hand object* | | | |
| EgoHOS (Zhang et al., 2022) | 0.562 | 19.022 | 868.928 |
| EgoHOS+SAM-2 | **0.571** | **19.212** | **836.033** |

**Analysis on data pipeline.** To verify the effect of our cross-view mask generation pipeline, we compare the video prediction model that is trained on HOI masks generated via EgoHOS only vs EgoHOS+SAM-2. Here, we apply the HOI masks extracted from future frames as conditions for straightforward comparison. EgoHOS (Zhang et al., 2022) performs per-frame hands and interacting object segmentation, while ignoring the temporal consistency across consequent frames. SAM-2 (Ravi et al., 2024) compensates the loss of temporal consistency by tracking hands and objects through all frames given mask prompts generated by EgoHOS. As seen in Table 6, SAM-2 leads to better generation performance due to the improved quality of both hand and object masks.

**Application to different diffusion models.** To demonstrate the generalization capability of our approach, we integrate our cross-view mask prediction model with different video diffusion models. By default, we adopt SEINE (Chen et al., 2023) as our primary video diffusion model. Additionally, we experiment with ConsistI2V (Ren et al., 2024) as an alternative diffusion model, and the training and inference pipeline remains unchanged. The experimental

Table 7: Generalisation ability of EgoExo-Gen to different video prediction models.

| Model | w/ exo | SSIM ↑ | PSNR ↑ | FVD ↓ |
|---|---|---|---|---|
| SEINE | ✗ | 0.518 | 17.680 | 1063.458 |
| SEINE | ✓ | **0.537** | **18.395** | **1031.693** |
| ConsistI2V | ✗ | 0.532 | 18.318 | 1109.314 |
| ConsistI2V | ✓ | **0.540** | **18.581** | **1017.377** |

results, as shown in Table 7, demonstrate that incorporating the cross-view mask prediction model leads to performance gains on both SEINE and ConsistI2V. This validates the generalization capability of our method in introducing cross-view information across different video diffusion models.

## 3.3 QUALITATIVE COMPARISON AND LIMITATIONS

We show the visualisation results of the predicted videos in Fig. 4. Our default model, *i.e.*, EgoExo-Gen (w/o future) predicts videos with reasonable hand-object movement, despite actions that require minor hand-object movement (*e.g.*, stirring). In comparison, the videos generated by ConsistI2V (Ren et al., 2024) could result in the unstable object shape (*e.g.*, bowl) or nearly static video in the $2^{nd}$ case. However, we find that both EgoExo-Gen and ConsistI2V fail in the case where complex hand movement is involved, as shown in the last row. In the $2^{nd}$ column, we also show the results of an oracle model, where our HOI-aware video diffusion model takes the hand-object masks extracted from the visible future frames using EgoHOS and SAM-2. Despite the aforementioned challenges, EgoExo-Gen (w/ future) predicts realistic videos that are close to the Ground-Truth.

**Text instruction:** C pours the eggs in the stainless bowl with his right hand.

**Text instruction:** C stirs the milk tea in the pot on the stovetop with the spoon in his right hand.

**Text instruction:** C adds the cut tomato to the bowl of salad mixture with his left hand.

Ground-Truth          EgoExo-Gen (w/ future) EgoExo-Gen (w/o future)          ConsistI2V

Figure 4: **Qualitative comparisons**. EgoExo-Gen (w/o future) refers to our default model using the predicted HOI masks as condition. EgoExo-Gen (w/ future) uses the HOI masks extracted from visible future frames (Sec: 2.4), serving as an oracle. The last row shows a failure case with complex hand movement. *Best viewed with Acrobat Reader. Click the image to view the animated videos.*

This indicates the HOI-aware video diffusion model in EgoExo-Gen learns good controllability over hands and objects, while the performance is bounded by the sub-optimal mask prediction ability, which can also be observed in Table 4 and Fig. 5. Improving the quality of predicted masks in future invisible frames (*e.g.*, $10^{th}$ frame) would be our future work.

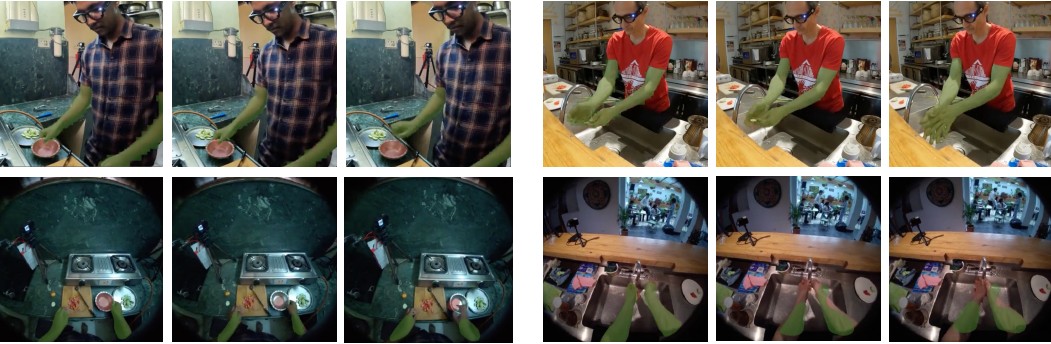

Figure 5: Visualisation of the exo-centric hand-object masks and predicted ego-centric masks at the visible $1^{st}$ frame and invisible $5^{th}$ and $10^{th}$ frames.

## 4 RELATED WORK

**Egocentric-Exocentric video understanding and generation.** Going beyond ego-only Wang et al. (2023b); Pei et al. (2024) or exo-only Wang et al. (2022) video analysis, video understanding from joint ego-centric and exo-centric perspectives covers a wide range of tasks, such as action recognition (Li et al., 2021; Xue & Grauman, 2023; Zhang et al., 2024), retrieval (Huang et al., 2024a; Xu et al., 2024), cross-view relation (Grauman et al., 2024), and skill assessment (Huang et al., 2024a; Li et al., 2024). While most works perform cross-view action understanding at video level, another line of research focuses on cross-view translation/generation (Regmi & Borji, 2018; Liu et al., 2021; Tang et al., 2019; Grauman et al., 2024; Liu et al., 2025), which requires building spatio-temporal relationship across different views. For instance, (Luo et al., 2024a) proposed to generate the exo-centric view by mining trajectory information from ego-centric view. (Luo et al., 2024b) realised exo-to-ego image generation by exploring hand pose as explicit guidance for an image diffusion model. In contrast to prior approaches, we design a cross-view mask prediction model to anticipate spatio-temporal masks of *both hand and interacting object* from exo-view to ego-view.

**Diffusion models for video prediction.** Video prediction (a.k.a, image animation) aims to generate the subsequent video frames given the first frame as condition (Xue et al., 2016; Franceschi et al., 2020; Voleti et al., 2022). Motivated by text-to-image (T2I) diffusion models (Rombach et al., 2022b; Saharia et al., 2022) and text-to-video (T2V) diffusion models (Zhou et al., 2022; Blattmann et al., 2023b; Guo et al., 2023b; Wu et al., 2023; Wang et al., 2023c; Ma et al., 2024), a line of works take the first frame as an extra condition to the T2V model (Chen et al., 2023; Zhang et al., 2023; Ren et al., 2024; Xing et al., 2023) to achieve controllable video prediction. ConsistI2V (Ren et al., 2024) conducted spatio-temporal attention over the first frame to maintain spatial and motion consistency. DynamiCrafter (Xing et al., 2023) designed a dual-stream image injection paradigm to improve the generation. In contrast to previous methods that have focused on generating videos in general domains, (Gu et al., 2023; Xing et al., 2024) target the prediction of real-world ego-centric videos by decomposing text instructions (Gu et al., 2023; Xing et al., 2024) or specialised adapter (Xing et al., 2024). Our work explicitly model the dynamics (*i.e.*, hands and interacting objects) in ego-centric video prediction.

**Hand-Object segmentation.** The analysis on Hand and object interaction (HOI) covers a wide range of research directions (Kumar et al., 2009; Shan et al., 2020; Xu et al., 2023b; Ohkawa et al., 2023a). Here, we focus on hand and interacting object segmentation (HOS), a challenging task that requires the model to segment open-world interacting objects Xu et al. (2023a); Wang et al. (2021); Ouyang et al. (2024). VISOR (Darkhalil et al., 2022) and EgoHOS (Zhang et al., 2022) proposed to segment left/right hands and interacting objects in ego-centric view. Despite excellent performance, these works generally fail in exo-centric view. To address HOS in third-person perspective, HOISTformer (Narasimhaswamy et al., 2024) designed a maskformer-based (Cheng et al., 2022) hand-object segmentation model trained on exo-centric dataset. Recent vision foundation models dedicated for segmentation (Khirodkar et al., 2024; Kirillov et al., 2023), SAM-2 (Ravi et al., 2024) is capable of segmenting and tracking objects given visual prompts. Sapiens (Khirodkar et al., 2024) excels at segmenting human body parts, including hands and arms. Our automated pipeline is built upon these works to segment hands and objects in both ego- and exo-view.

## 5 CONCLUSION

In this paper, we propose EgoExo-Gen to solve cross-view video prediction task by modeling hand-object dynamics in the video. EgoExo-Gen combines (1) a cross-view mask prediction model that estimates hand-object masks in unobserved ego-frames by modeling the spatio-temporal ego-exo correspondence and (2) a HOI-aware video diffusion model that incorporates the predicted HOI masks as structural guidance. We also devise an automated HOI mask annotation pipeline to generate HOI masks for both ego- and exo-videos, enhancing the scalability of EgoExo-Gen. Experiments demonstrate the superiority of EgoExo-Gen over prior video prediction models in predicting videos with realistic hand-object movement, revealing its potential application in AR/VR and embodied AI.

**Acknowledgement** This work is supported by Natural Science Foundation of China (No. 62172101), and the Science and Technology Commission of Shanghai Municipality (No. 24511103301).

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

# A    ADDITIONAL EXPERIMENTS

## A.1    NUMBER OF CONDITIONING HOI FRAMES

First, we investigate the impact of varying the number of given HOI mask frames on ego-centric video prediction. The baseline model is trained without HOI mask condition. When altering the number of HOI conditioning frames, the remaining HOI masks are set to zero. As depicted in the Fig. 6 top, EgoExo-Gen significantly outperforms the baseline by utilising only one or two HOI conditioning frames. As the number of HOI mask frames gradually increases, EgoExo-Gen's performance steadily improves while approaching saturation around 12 frames, due to better encoding of hand and object dynamics. Next, we compare model performance as we continuously increase the number of beginning RGB frames as conditions. Notably, our default setting uses only the first RGB frame as condition. Fig. 6 bottom shows that both EgoExo-Gen and the baseline model significantly improve as the frame number increases, thanks to the rich visual context provided by the beginning frames. In this case, EgoExo-

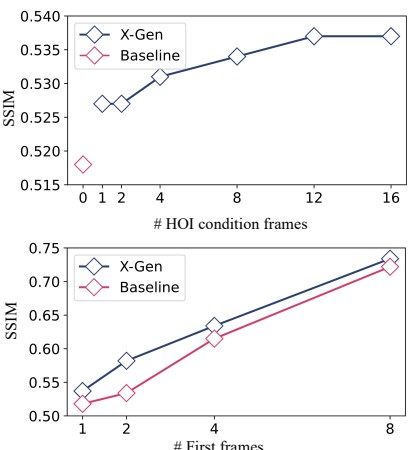

Figure 6: Comparison on the number of HOI conditioning frames (top) and RGB conditioning frames (bottom).

Gen still outperforms the baseline on all numbers of conditioning frames, indicating that subsequent HOI masks also contribute to the prediction of future frames.

## A.2    DYNAMICS THE MODEL IS LEARNING

Given that the average duration of video clips in Ego-Exo4D is relatively short (~1 second), we conduct experiments to validate the dynamics the model is learning in two aspects.

**Dynamics of the dataset.** We first analyse the dynamics within the Ego-Exo4D dataset. Specifically, for each video clip, we uniformly sample 16 frames and calculate the Intersection over Union (IoU) between the hands and interactive objects in the first frame and subsequent frames. We select frames 2, 4, 8, 12, and 16 for this analysis. Results in Table 8 show that the IoU between the first frame and subsequent frames gradually decreases, indicating the spatial dynamic changes of hand and interactive objects.

Table 8: Mask IoU between the first frame and subsequent frames.

| Class/Frame | 2 | 4 | 8 | 12 | 16 |
|---|---|---|---|---|---|
| Hand | 0.66 | 0.45 | 0.31 | 0.25 | 0.23 |
| Interacting object | 0.67 | 0.48 | 0.33 | 0.27 | 0.24 |

**Dynamics the model learns.** Next, we compare the model's performance on videos with different dynamics. In particular, we calculate the averaged optical flow of the videos in the validation set using RAFT [4], and split the videos into two sets using a threshold, i.e., small-flow (SF) set and large-flow (LF) set. As shown in Table 9, compared to the baseline model, our model achieves more significant improvements on the large-flow set, demonstrating that our HOI condition effectively helps the model generate videos with dynamics. Despite the improvement, our model struggles to

predict reliable future ego-centric frames where complex hand-object movement or rapid camera movement is involved, as shown in the last row of Fig. 4.

Table 9: Comparison on the model's performance on small-flow (SF) and large-flow (LF) subsets.

| Methods | SSIM (SF) | LPIPS (SF) | SSIM (LF) | LPIPS (LF) |
|---------|-----------|------------|-----------|------------|
| Baseline | 0.546 | 0.259 | 0.470 | 0.381 |
| Ours | 0.565 | 0.238 | 0.512 | 0.354 |

## A.3 HOI MASK VS. ALL-OBJECT MASK

In egocentric videos, the primary motion is centered around hand-object interactions, which play a crucial role in tasks such as pose estimation (Kwon et al., 2021; Ohkawa et al., 2023b), action recognition and segmentation (Damen et al., 2018; Huang et al., 2020b;a; Chen et al., 2024), interaction anticipation (Grauman et al., 2022), and hand motion trajectory prediction. To justify the need of hand-object masks in predicting ego-centric videos, we also compare our HOI mask with all-object mask. In particular, we adopt SAM-2 (Ravi et al., 2024) to segment and track every possible object without considering their classes, and train the mask encoder using this all-object masks. Note that we do not train a cross-view mask prediction with this data as the correspondence between exo-objects and ego-objects is not available due to the class-agnostic nature of SAM-2. Therefore, at inference time, the all-object mask is only available for the first frame. As observed in Table 10, using all-object masks does not lead to better performance, primarily because it is limited to masks from only the first frame, which fail to provide accurate motion guidance. In contrast, our model benefits from cross-view mask prediction, enabling the prediction of HOI masks for all frames. In future work, we aim to extend this approach to include a broader range of objects that are relevant to the ongoing task.

Table 10: Comparison between HOI mask and all-object mask

| Conditions | SSIM | PSNR | LPIPS | FVD |
|------------|------|------|-------|-----|
| All-object mask (first frame) | 0.478 | 16.681 | 0.420 | 1324.360 |
| HOI mask (first frame) | 0.529 | 18.143 | 0.322 | 1096.771 |
| HOI mask (ours, all frames) | 0.537 | 18.395 | 0.311 | 1031.693 |

## A.4 BALANCE BETWEEN EGO AND EXO MEMORY ATTENTION

In the ego-exo memory attention module, the final feature is calculated as $Z'' = \alpha Z' + (1 - \alpha)Z$, where $Z'$ and $Z$ denotes ego and exo features, respectively. At training stage, we adopt a step decay mechanism for $\alpha$, which is represented as:

$$
\alpha(t) = \begin{cases}
1.0, & 0 \leq t < 0.5T, \\
0.8, & 0.5T \leq t < 0.6T, \\
0.6, & 0.6T \leq t < 0.7T, \\
0.4, & 0.7T \leq t < 0.8T, \\
0.2, & 0.8T \leq t < 0.9T, \\
0.0, & 0.9T \leq t < T,
\end{cases}
$$

where $t$ and $T$ refer to the current training iteration and total iterations, respectively.

Table 4 in our manuscript compares our approach with methods that use (1) ego feature only (i.e. $\alpha$=1.0) and (2) exo feature only (i.e. $\alpha$=0.0). Here, we also add a comparison that adopts cosine decay. As list in Table 11, cosine decay performs worse than step decay, possibly because introducing exo features too early during the training phase might interfere with the model's fundamental segmentation ability. In contrast, step decay strategy helps the model training in the early stage, and eventually learns to predict the mask features for all egocentric frames at inference time.

Table 11: Comparison of different decay strategies.

| Methods | IoU | CA | LE | SSIM | PSNR | LPIPS | FVD |
|---------|-----|-----|-----|------|------|-------|-----|
| Cosine decay | 13.632 | 0.132 | 0.131 | 0.530 | 18.131 | 0.321 | 1115.105 |
| Step decay | 20.315 | 0.207 | 0.082 | 0.537 | 18.395 | 0.311 | 1031.693 |

