# OpenReview forum: "EgoExo-Gen: Ego-centric Video Prediction by Watching Exo-centric Videos"
_ICLR.cc/2025/Conference — ICLR 2025 Poster_

### Official Review · Reviewer_yjuR · 2024-11-02

**Soundness:** 3
**Presentation:** 3
**Contribution:** 2
**Rating:** 6
**Confidence:** 4

**Summary:**

- The paper addresses cross-view video prediction, where the goal is to animate an ego-centric video starting from a single frame and guided by a corresponding exo-centric video and textual commands.
- The paper introduces an "ego-exo memory attention" mechanism that enhances the ability to transfer relevant features from exo-centric to ego-centric frames, aiding in the accurate prediction of interactions.
The proposed model is evaluated on Ego-Exo4D and H2O and shows superior performance over previous models, particularly in generating realistic hand and object interactions in ego-centric videos.

**Strengths:**

- X-Gen effectively leverages information from exo-centric videos to predict ego-centric video frames. This innovative approach bridges the gap between different perspectives, using third-person videos to enhance first-person video prediction.
- The paper introduces a novel approach to predict hand-object interaction (HOI) masks in future frames, which is critical for accurately generating frames that involve interactions with objects.
- The fully automated pipeline for generating HOI masks using vision foundation models reduces the reliance on manual annotations and increases the scalability of the training process.
- X-Gen demonstrates strong zero-shot transfer capabilities, performing well on unseen actions and environments in benchmark datasets.

**Weaknesses:**

See the questions below.

**Questions:**

- What were the key factors that influenced the architectural design of the X-Gen model, particularly the integration of the cross-view HOI mask prediction with the video diffusion process?
- Can you discuss specific instances where X-Gen failed to predict accurate video frames?
- Can you provide more detail on how the HOI mask prediction model handles the temporal dynamics and variability in human-object interactions across different video frames?
- What are the computational performances for training and testing the X-Gen mode?

---

> ### Author Response · Authors · 2024-11-23
> **(1/2) Response to Reviewer yjuR**
>
> Thank you for recognising our work and the valuable comments.
> We hope that the following point-to-point responses address your concerns, and that you could increase the rating accordingly.
>
> ## **Q1.Key factors that influenced the architectural design of the X-Gen model.**
>
> **A1.1 Key factors of the HOI-aware video diffusion model.**
> The considered task in this paper is cross-view video prediction, which requires the video prediction model to generate egocentric video that
> - aligns with the environmental context given by the first frame condition and maintains the temporal consistency of the generated video. This is modeled by temporal attention layers.
> - follows the given text instruction, which is modeled by cross attention layers.
> - aligns with the hand-object motion presented in exocentric video. The translated ego HOI mask serves as additional input condition to the video diffusion model.
>
> **Ablation experiments in Table 3** in our manuscript validate the effectiveness of each modality in generating reliable egocentric videos.
>
> **A1.2. Key factors of the cross-view mask prediction model.**
> The cross-view mask prediction model is constructed to translate the hand-object motion from exo view to ego view. The key contribution is the **ego-exo memory attention block**, which aims to
> - model the fine-grained, temporal relationship of ego/exo videos
> - leverage exocentric hand-object clues to infer egocentric hand-object mask features
>
> **Ablation experiments in Table 4** in our manuscript show the importance of ego-exo memory attention in improving both segmentation and generation.
>
> **A1.3. The integration of two models.**
> In the X-Gen model, the core question connecting two models is: **"What information from the exocentric perspective is critical in assisting egocentric video prediction?"**
> In this work, we choose to model **hand-object dynamics**, as hands and interactive objects in ego view reflect the user's intentions and characterize ongoing actions.
> Hand-object interactions have also received significant attention in previous egocentric research studies, including pose estimation [1], action recognition [2], interaction anticipation [3], hand motion trajectory prediction [4].
>
> **Ablation experiments in Tables 2 and 5** in our manuscript reveal that
> - both hand and objects clues are critical in assisting egocentric video prediction.
> - HOI masks serve as a better option in bridging exocentric and egocentric views.
>
> [1] Kwon, Taein, et al. "H2o: Two hands manipulating objects for first person interaction recognition." CVPR 2021.
>
> [2] Damen, Dima, et al. "Rescaling egocentric vision: Collection, pipeline and challenges for epic-kitchens-100." IJCV 2022.
>
> [3] Grauman, Kristen, et al. "Ego4d: Around the world in 3,000 hours of egocentric video." CVPR 2022.
>
> [4] Zhan, Xinyu, et al. "OAKINK2: A Dataset of Bimanual Hands-Object Manipulation in Complex Task Completion." CVPR 2024.
>
> ## **Q2.Failure cases of X-Gen.**
>
> **A2.1 Complex hand-object movement.**
> In the third case (last row) of Figure 5 in our manuscript, we provide the visualization results of a failure case where complex hand movement is involved in the action, i.e., **C adds the cut tomato to the bowl of salad mixture with his left hand.** Complex hand-object motion poses additional challenges for the cross-view mask prediction model to make reliable HOI masks.
>
> **A2.2 Camera movement.**
> Another challenge is the potential camera movement in the video. For example, some actions involve head movements, such as retrieving an avocado from a refrigerator behind.
> These scenarios often result in rapid scene transitions and potential video blurriness, making it challenging for generative models to accurately predict motion.
> Naive solutions would include:
> - filtering videos containing rapid camera movement by estimating the optical flow [1] of the videos.
> - controling the video prediction model with camera poses [2].
>
> We will add analyses on such failure cases in our revised manuscript.
>
> [1] Teed, Zachary, and Jia Deng. "Raft: Recurrent all-pairs field transforms for optical flow." ECCV 2020.
>
> [2] He, Hao, et al. "Cameractrl: Enabling camera control for text-to-video generation." Arxiv 2024.

---

> ### Author Response · Authors · 2024-11-23
> **(2/2) Response to Reviewer yjuR**
>
> ## **Q3.Detail on how the HOI mask prediction model handles the temporal dynamics and variability.**
>
> **A3.**
> The key module in our cross-view mask prediction model, i.e., ego-exo memory attention, is designed to model such temporal dynamics and variability in hand-object interactions.
>
> Specifically, we use Eq.(2) in our manuscript to illustrate the process:
> $$\begin{aligned}
> \mathcal{Q}^{\text{exo}}&=W _ q^{\mathrm{T}}\overline{x} _ n, \\\\
> \mathcal{K}&=W _ k^{\mathrm{T}}[\overline{x} _ 1,\overline{g} _ 1,\dots,\overline{x} _ {n-1},\overline{g} _ {n-1}], \\\\
> \mathcal{V}&=W _ v^{\mathrm{T}}[\overline{m}^x _ 1,\hat{\overline{m}}^g _ 1,\dots,\overline{m}^x _ {n-1},\hat{\overline{m}}^g _ {n-1}], \\\\
> \end{aligned}$$
> where $\{\overline{x} _ 1,...,\overline{x} _ {n-1}\}$ and $\{\overline{g} _ 1,...,\overline{g} _ {n-1}\}$ refer to historical exo/ego image features from frame 1 to $n$-1; $\{\overline{m} _ 1^{x},...,\overline{m} _ {n-1}^{x}\}$ and $\{\hat{\overline{m}} _ 1^{g},...,\hat{\overline{m}} _ {n-1}^{g}\}$ denote historical exo/ego mask features.
>
> Given the current exocentric video feature $\overline{x}_{n}$ of frame $n$, the attention operation first calculates $\mathcal{A} = \text{softmax}(\mathcal{Q}^{exo}\mathcal{K}^{\mathrm{T}})\in\mathbb{R}^{2(n-1)}$.
> Here, $\mathcal{A}$ reflects the normalised visual similarity of current frame and historical exo/ego frames.
> Then, the mask features are fused based on the visual similarity via $\mathcal{A}^{\mathrm{T}}\mathcal{V}$.
>
> Despite potential variability in HOI, the attention mechanism over the temporal dimension enables the model to find relevant visual ego/exo clues in historical frames and leverage their corresponding mask features to assist the mask prediction of the current frame.
> As the ego-exo memory attention is operated iteratively, the image/mask features of the current frame $n$ are also stored in the memory bank, (i.e., **preserving hand-object dynamics and variability as much as possible**) to facilitate the mask prediction in future frames.
>
>
> ## **Q4.Computational performances for training and testing.**
>
> **A4.**
> In our understanding, the **computational performances** here refers to the training/inference computational costs. Results can be found in the following Table.
> We report results of both the cross-view mask prediction model (Seg) and video prediction model (Gen).
> Regarding the Gen model, we list the computational costs for the backbone (stage-1) and the mask encoder (stage-2).
> During training, we use 8 A100 GPUs with a batch size of 32.
> At inference time, we employ a single A100 GPU and evaluate the model with a batch size of 1.
>
> | Methods    | Trainable params  (MB) | Training time  (sec/iter) | Inference speed (FPS) | Memory (GB) |
> | ----------- | ----------- | ----------- | ----------- | ----------- |
> | Seg model | 62 | 3.1  | 30   | 61    |
> | Gen model (stage-1) | 909 | 1.5  |1.1   | 66   |
> | Gen model (stage-2) | 277 | 1.3 | 0.9 | 52 |

---

> > ### Comment · Reviewer_yjuR · 2024-11-24
> >
> > The author's response has addressed my concerns; I will increase the score from 5 to 6.

---

### Official Review · Reviewer_5j2P · 2024-11-04

**Soundness:** 2
**Presentation:** 3
**Contribution:** 2
**Rating:** 6
**Confidence:** 4

**Summary:**

The paper aims to generate the ego-centric videos given the first frame of the ego-centric video, a text instruction, and a synchronized exo-centric video. The proposed model, X-Gen, involves two components: i) an exo-to-ego HOI mask prediction framework, and ii) an ego-centric video diffusion model given the first frame of the ego view, the text instruction, and the predicted ego HOI mask from the first component. Experiments were mainly conducted with the Ego-Exo4D dataset, where the authors adopted off-the-shelf models (e.g. EgoHOS, SAM2) to generate HOI mask annotations. The zero-shot performance of X-Gen was also evaluated with the H2O dataset.

**Strengths:**

1.	The manuscript is well organized in general.
2.	The paper introduces high-level novelty on using cross-view HOI mask prediction to guide the video diffusion model.
3.	Several ablations and visualizations were shown in the experiment section.

**Weaknesses:**

1.	The justification for the need of the proposed cross-view mask prediction network is not strong. For example, given that Ego video frames are available during training, one baseline can be using a HOI mask predictor for only the Ego views, either with off-the-shelf HOI detectors (e.g. EgoHOS+SAM2) or training one with the dataset. Another can be using Exo-Ego video frames without Exo HOI mask.

2.	Given that the average video duration is only 1 second (L268), it is unclear how much dynamics the model is learning. What are the evaluation metrics in Table 1 if only the HOI mask of the first Ego video frame is used as the condition? Also, what about using the first Ego video frame directly as the predictions?

3.	It is unclear why prior cross-view transformation modules (e.g. [a]) cannot be used as a baseline for the first component.

4.	In L269, the authors claimed that the object masks from the cross-view relation benchmark are not guaranteed to be interacting objects, and the hand masks are not annotated. However, there is no justification that using HOI masks is a better option.

5.	It is unclear how alpha was annealed during training (L175) and there is no experiment showing that whether it is important.

**Ref**:

-	[a] Yang et al., Projecting Your View Attentively: Monocular Road Scene Layout Estimation via Cross-view Transformation, CVPR 2021.

**Questions:**

In addition to the questions in the weakness section,

1.	Can the authors provide more details about the pipeline at the inference time? E.g., what are the inputs? What will Eq. (2) turn to? Do you take the predicted Ego frames from the second component to the first component, why or why not?
2.	Can the authors elaborate more details about the temporal attention blocks in the video diffusion part? How was the temporal information fused here?
3.	What does it mean by “we apply the hand-object masks extracted from the future video frames instead of cross-view mask predictions” in Table 2 and 3? Are they ground truth HOI masks?

---

> ### Author Response · Authors · 2024-11-23
> **(1/4) Response to Reviewer 5j2P**
>
> Thank you for recognising our work and the valuable comments.
> We hope that the following point-to-point responses address your concerns, and that you could increase the rating accordingly.
>
> ## **Q1. Using off-the-self HOI detectors or Exo-Ego video frames without Exo HOI mask.**
>
>
> **A1.1 Off-the-shelf HOI detectors.** We agree that using off-the-shelf HOI detectors (e.g. EgoHOS+SAM2) for producing HOI masks is a good option while training the HOI-aware video diffusion model, and this is **exactly our training scheme to ensure the mask quality (Line 254-255)**.
> However, off-the-shelf HOI detectors cannot work at inference time as only the first ego-frame is visible, meaning that only the HOI mask of the first frame can be obtained.
> We compare X-Gen with the method that only adopts the HOI mask of the first frame in the Table below. Using only the HOI mask from the first frame as a condition improves SSIM and PSNR compared to the baseline (No mask).
> However, it does not lead to improvements in LPIPS and FVD. In contrast, our proposed cross-view mask predictions achieve improvements across all metrics.
>
> | Methods    | SSIM $\uparrow$ | PSNR $\uparrow$ | LPIPS $\downarrow$ | FVD $\downarrow$ |
> | ----------- | ----------- | ----------- | ----------- | ----------- |
> | No mask | 0.518 | 17.680  | 0.321   | 1063.458    |
> | HOI mask (first frame) | 0.529 | 18.143  | 0.322   | 1096.771  |
> | HOI mask (ours, all frames)  | **0.537** | **18.395**  | **0.311**   | **1031.693**  |
>
>
> **A1.2 Exo-Ego frames without HOI mask.**
> In Table 5 in our manuscript, we compare different design choices of incorporating exocentric information. Among the choices, one approach is replacing the predicted Ego HOI masks with Exo RGB frames, without introducing HOI masks in the entire process.
> The sub-optimal performance of Exo RGB frame condition indicates the difficulty of learning exo-ego translation and video prediction in a single video diffusion model.
> In contrast, HOI masks in the ego-view provide explicit pixel-aligned visual clues to improve the video prediction.
> We will implement more baseline approaches (e.g. Pix2PixHD[1], Vid2Vid[2], Pix2Pix-Turbo[3]) for comprehensive evaulation.
>
> | Conditions    | SSIM $\uparrow$ | PSNR $\uparrow$ | LPIPS $\downarrow$ |FVD $\downarrow$ |
> | ----------- | ----------- | ----------- | ----------- | ----------- |
> | Exo RGB frames | 0.525 | 17.901  | 0.338  | 1094.316  |
> | HOI mask (ours, all frames)  | **0.537** | **18.395**  | **0.311**   | **1031.693**  |
>
> [1] Wang, Ting-Chun, et al. "High-resolution image synthesis and semantic manipulation with conditional gans." CVPR 2018.
>
> [2] Wang, Ting-Chun, et al. "Video-to-video synthesis." ArXiv 2018.
>
> [3] Parmar, Gaurav, et al. "One-step image translation with text-to-image models." Arxiv 2024.

---

> ### Author Response · Authors · 2024-11-23
> **(2/4) Response to Reviewer 5j2P**
>
> ## **Q2. The dynamics the model is learning / the performance of only using the HOI condition of first Ego video frame / using the first Ego video frame directly as the predictions**
>
> **A2.1 HOI mask of the first frame.**
> We include the additional results of only using the HOI mask of the first frame.
> Our approach achieves superior performance than only using the HOI mask of the first frame, indicating the HOI masks in subsequent frames also provide valuable structual guidence.
>
> **A2.2 Using first ego frame as the predictions.**
> We replace the predicted first frame with the ground-truth frame, and we observe no significant performance change.
> This indicates that the diffusion model effectively leverages the first-frame condition by predicting plausible first frame that closely aligns with the ground truth.
>
> | Conditions    | SSIM $\uparrow$ | PSNR $\uparrow$ | LPIPS $\downarrow$ |FVD $\downarrow$ |
> | ----------- | ----------- | ----------- | ----------- | ----------- |
> | First frame mask   | 0.529 | 18.143  | 0.322   | 1096.771  |
> | First frame as GT  | **0.538** | **18.400**  | 0.312   | **1030.225**  |
> | Ours               | 0.537 | 18.395  | **0.311**   | 1031.693  |
>
>
> **A2.3 How much dynamics the model is learning.**
> To answer this question, we first analyse the dynamics within the dataset.
> Specifically, for each video clip, we uniformly sample 16 frames and calculate the Intersection over Union (IoU) between the hands and interactive objects in the first frame and subsequent frames. We select frames 2, 4, 8, 12, and 16 for this analysis. Results in the table below show that the IoU between the first frame and subsequent frames gradually decreases, indicating the spatial dynamic changes of hand and interactive objects.
>
> | Class/Frame    | 2 | 4 | 8 | 12 | 16 |
> | ----------- | ----------- | ----------- | ----------- |----------- |----------- |
> | Hand | 0.66 | 0.45 | 0.31 | 0.25 | 0.23 |
> | Object | 0.67 | 0.48 | 0.33 | 0.27 | 0.24 |
>
>
> Next, we compare the model's performance on videos with different dynamics. In particular, we calculate the averaged optical flow of the videos in the validation set using RAFT [4], and split the videos into two sets using a threshold, i.e., small-flow (SF) set and large-flow (LF) set.
> Compared to the baseline model, our model achieves more significant improvements on the large-flow set, demonstrating that our HOI condition effectively helps the model generate videos with dynamics.
>
> | Methods | SSIM (SF) | LPIPS (SF) | SSIM (LF) | LPIPS (LF) |
> | ----------- | ----------- | ----------- | ----------- |----------- |
> | Baseline | 0.546 | 0.259 | 0.470 | 0.381 |
> | Ours | **0.565** | **0.238** | **0.512** | **0.354** |
>
> [4] Teed, Zachary, and Jia Deng. "Raft: Recurrent all-pairs field transforms for optical flow." ECCV 2020.
>
> ## **Q3. Using cross-view tranformation [5] as a baseline.**
>
> **A3.**
> This work [5] focuses on the autonomous driving domain, which presents a significant gap compared to our scenario involving indoor exo-ego setups with a focus on human activity.
> In [5], the cross-view transformation module predicts the vehicle occupancy in the bird's-eye view (BEV) from the front-view monocular image.
> Despite the domain gap, we implement this approach on our exo-ego data. Here, we adopt two variants with different inputs:
> - **Variant-A**: The input is the exocentric RGB frame and the output is the corresponding ego HOI mask;
> - **Variant-B**: The input is the channel-wise concatenation of the exo frame and exo HOI mask and the output remains the ego HOI mask.
>
> We train the model for 30 epochs with a batch size of 64 on 1*A100 GPU.
> We take the centre frame of the video clip/HOI video mask as the input/output.
> The results listed below reveal that this approach is not suitable for our cross-view mask prediction task.
> We hypothesize two reasons for this discrepancy:
> - The model is image-based and lacks specific design for temporal modeling, whereas our approach incorporates temporal memory attention.
> - [5] primarily focuses on the translation of single category (e.g., vehicles), while our setting involves open-world interactive objects. In our scenario, the model is required to infer ego HOI masks from exo HOI masks, posing additional challenges. Notably, we conduct an experiment by introducing exo HOI masks (Variant-B), which exhibits improvement over Variant-A. However, it still falls short compared to our method.
>
>
> | Methods    | IoU $\uparrow$ | CA $\uparrow$ | LE $\downarrow$ |
> | ----------- | ----------- | ----------- | ----------- |
> | Variant-A | 2.3 | 0.053  | 0.201 |
> | Variant-B | 2.7 | 0.069  | 0.187 |
> | Ours      | **20.3** | **0.207**  | **0.082** |
>
> [5] Yang et al., Projecting Your View Attentively: Monocular Road Scene Layout Estimation via Cross-view Transformation, CVPR 2021.

---

> ### Author Response · Authors · 2024-11-23
> **(3/4) Response to Reviewer 5j2P**
>
> ## **Q4.Need Justification that using HOI masks is a better option.**
>
> **A4.**
> In egocentric videos, the primary motion is centered around hand-object interactions, which play a crucial role in tasks such as pose estimation [6], action recognition [7], interaction anticipation [8], hand motion trajectory prediction [9].
>
> In addition, in the Ego-Exo4D dataset, each ego camera is paired with at least four exo cameras, ensuring that hand-object interactions are generally visible in both the ego and at least one exo perspective. Other objects, however, may not be visible in the ego/exo view.
>
> In comparison, we adopt SAM-2 to segment and track every possible object without considering their classes, and train the mask encoder using this all-object masks.
> Note that we do not train a cross-view mask prediction with this data as the correspondence between exo-objects and ego-objects is not available due to the class-agnostic nature of SAM-2.
> Therefore, at inference time, the all-object mask is only available for the first frame.
> As observed in the below Table, using all-object masks does not lead to better performance, primarily because it is limited to masks from only the first frame, which fail to provide accurate motion guidance.
> Our model benefits from cross-view mask prediction, enabling the prediction of HOI masks for all frames.
> In future work, we aim to extend this approach to include a broader range of objects.
>
> | Conditions    | SSIM $\uparrow$ | PSNR $\uparrow$ | LPIPS $\downarrow$ |FVD $\downarrow$ |
> | ----------- | ----------- | ----------- | ----------- | ----------- |
> | All-object mask (first frame) | 0.478 | 16.681  | 0.420  | 1324.360 |
> | HOI mask (first frame)  | 0.529 | 18.143  | 0.322   | 1096.771  |
> | HOI mask (ours, all frames) | **0.537** | **18.395**  | **0.311**   | **1031.693**  |
>
>
> [6] Kwon, Taein, et al. "H2o: Two hands manipulating objects for first person interaction recognition." CVPR 2021.
>
> [7] Damen, Dima, et al. "Rescaling egocentric vision: Collection, pipeline and challenges for epic-kitchens-100." IJCV 2022.
>
> [8] Grauman, Kristen, et al. "Ego4d: Around the world in 3,000 hours of egocentric video." CVPR 2022.
>
> [9] Zhan, Xinyu, et al. "OAKINK2: A Dataset of Bimanual Hands-Object Manipulation in Complex Task Completion." CVPR 2024.
>
> ## **Q5. How alpha was annealed during training.**
>
> **A5.**
> Recall that our feature $Z''=\alpha*Z'+(1-\alpha)*Z$, where $Z'$ and $Z$ denotes ego and exo features, respectively. At training stage, we adopt a step decay mechansim for $\alpha$ representated as:
>
> $$
> \alpha = \begin{cases}
> 1.0, & 0 \leq t < 0.5T \\\\
> 0.8, & 0.5T \leq t < 0.6T \\\\
> 0.6, & 0.6T \leq t < 0.7T \\\\
> 0.4, & 0.7T \leq t < 0.8T \\\\
> 0.2, & 0.8T \leq t < 0.9T \\\\
> 0.0, & 0.9T \leq t < T \\\\
> \end{cases}
> $$
>
> where $t$ and $T$ refers to the current iteration and total training iteration, respectively. Such strategy helps the model training in the early stage, and eventually learns to predict the mask features for all egocentric frames at inference time.
>
> Table 4 in our manuscript compares our approach with methods that use (1) ego feature only (i.e. $\alpha$=1.0) and (2) exo feature only (i.e. $\alpha$=0.0), here, we also add a comparison that adopts cosine decay for $\alpha$.
> The results show that cosine decay performs worse than step decay, possibly because introducing exo features too early during the training phase might interfere with the model's fundamental segmentation ability.
> We will add this result on the final paper.
>
> | Methods    | IoU $\uparrow$ | CA $\uparrow$ | LE $\downarrow$ | SSIM $\uparrow$ | PSNR $\uparrow$ | LPIPS $\downarrow$ |FVD $\downarrow$ |
> | ----------- | ----------- | ----------- | ----------- |  ----------- | ----------- | ----------- | ----------- |
> | Cosine decay | 13.632 | 0.132 | 0.131 | 0.530 | 18.131 | 0.321 | 1115.105 |
> | Step decay (ours) | **20.315** | **0.207** | **0.082** | **0.537** | **18.395** | **0.311** | **1031.693** |

---

> ### Author Response · Authors · 2024-11-23
> **(4/4) Response to Reviewer 5j2P**
>
> ## **Q6. Details about the pipeline at the inference time? / Iterative processing of two components.**
>
> **A6.1.Inference pipeline of the cross-view mask prediction model.**
> At inference time, the inputs to the cross-view mask prediction model include
> - the exocentric video $\mathcal{V} _ {exo}$ with corresponding exocentric HOI mask $\mathcal{M}_{exo}$.
> - the egocentric video $\mathcal{V}_{ego}$ (with $2^{nd}$ frame to $N^{th}$ frame set to zero image)
>
> The model predicts the HOI mask of all egocentric frames $\hat{\mathcal{M}}_{ego}$ all at once.
> In the memory attention block, Eq.(2) remains the same. Here, the key is defined as:
> $$
> \mathcal{K}=W _ k^{\mathrm{T}}[\overline{x} _ 1,\overline{g} _ 1,\dots,\overline{x} _ {n-1},\overline{g} _ {n-1}]
> $$
>
> where $\overline{g}_1$ represents the feature of the visible $1^{st}$ ego frame, and $\overline{g} _ 2,...,\overline{g} _ {n-1}$ refer to the zero image feature. Hence, the model is required to additionally leverage useful exocentric clues to make predictions.
>
> **A6.2.Inference pipeline of the video prediction model.**
> The models takes as input:
> - The $1^{st}$ egocentric frame $g_1$
> - The predicted mask sequence of the ego video $\hat{\mathcal{M}}_{ego}$
> - Text instruction $\mathcal{T}$
>
> and outputs the predicted ego RGB frames $\{\hat{g}_1,...,\hat{g}_N\}$ all at once.
>
> **A6.3.Iterative processing of mask segmentation and video generation.**
> Thanks for your suggestion. This is a very good choice to improve the quality of both the ego-mask and the ego-video.
> Currently, we do not take the predicted ego frames back to the segmentation model because the base video diffusion model (i.e. SEINE) is designed to predict all frames (16 frames in our case) all at once by leveraging temporal attention.
> Iterative processing would involve further investigation on the architecture and perhaps re-training of the video diffusion model, and it is challenging to re-implement this during the rebuttal period.
> We consider this as a valuable future direction to improve our work.
>
> ## **Q7.Details about the temporal attention blocks.**
>
> **A7.**
> The temporal attention in the video diffusion model takes as input the hidden video features $F\in\mathbb{R}^{b\times {hw}\times f\times d}$ where $b$, $hw$ and $f$, $d$ refers to the batch size, spatial resolution, number of frames and hidden dimension, respectively.
> $F$ is then reshaped to $(b\times hw) \times f\times d$. A standard attention mechanism is then operated over the temporal dimension, i.e., modeling the temporal relationship among frames.
> The output feature has the same dimension as the input feature.
>
> ## **Q8.Explanation on HOI masks extracted from the future video frames.**
>
> **A8.**
> Yes, we choose GT HOI masks to guarantee the mask quality for merely evaluating the video prediction model.

---

> ### Comment · Reviewer_5j2P · 2024-11-25
> **Thanks for the rebuttal**
>
> Thank the authors for providing more details and ablations. The rebuttal addressed my concerns.

---

### Official Review · Reviewer_JxPc · 2024-11-04

**Soundness:** 3
**Presentation:** 3
**Contribution:** 2
**Rating:** 8
**Confidence:** 3

**Summary:**

This paper proposes a novel approach ( X-Gen) for generating future frames in ego-centric videos based on exo-centric footage and textual instructions. By modeling hand-object interactions (HOI) and employing a two-stage process that predicts HOI masks and utilizes a video diffusion model, X-Gen enhances prediction quality. Extensive experiments show that X-Gen outperforms existing models, particularly in generating realistic hand and object interactions.

**Strengths:**

--> The paper is well-written and easy to understand. All the key contributions are clearly presented with individual sections describing the components of the model in detail.
--> Experimental evaluation is thorough with a detailed ablation study. These experiments clearly show the impact of cross-view HOI mask prediction on the overall performance.
--> The automated approach to generate Ego-Exo HOI masks is also a good contribution.

**Weaknesses:**

--> ConsistI2V trained on Ego-Exo4D achieves SSIM of 0.532, compared to X-Gen which achieves 0.537. The difference is not significant. Also ConsistI2V only need the first frame (in ego view) and the text to generate the output, whereas X-Gen would also need the entire exo video and have to perform cross-view HOI mask prediction to generate the output. Given the overhead and the additional requirements of X-Gen, along with the marignal improvement in performance,  the novelty and adoption of this method are called into question.
--> Adding the details about the training time, inference time, number of trainable parameters and the compute resources required for training would improve the paper.

**Questions:**

--> Inputs to your model are the exo video, first frame of the corresponding ego video and the textual description. How will your approach perform if the inputs are the exo video and the first frame of a random ego video and the textual description? If the correspondence between the exo video and ego frame is required, then what is the need usecase where this method will be useful.

---

> ### Author Response · Authors · 2024-11-23
> **(1/2) Response to Reviewer JxPc**
>
> Thank you for recognising our work and the valuable comments.
> We hope that the following point-to-point responses address your concerns, and that you could increase the rating accordingly.
>
> ## **Q1. Improvement over ConsistI2V and the adoption of HOI condition in X-Gen**
>
> **A1.1 The adoption of HOI condition.**
>
> X-Gen is proposed to introduce exocentric videos to provide structural guidance for generating reliable hand-object interaction motions in video prediction tasks, whereas relying solely on the first frame and text as guidance lacks the crucial video-level motion cues necessary for generating realistic predictions.
> The effectiveness of HOI condition is validated by ablation experiments in Table 3 in our manuscript, and the conclusions include:
> - **ID1 vs. ID2, ID3 vs. ID4**, introducing HOI condition significantly boost the generation performance w/ or w/o text instructions.
> - **ID2 vs. ID3**, structural HOI control of the generation appears more effective than text instructions, as the hand-object movement can also serve as a valuable cue for inferring the current action semantics.
>
>
> | Exp_ID | Conditions      | SSIM $\uparrow$ | PSNR $\uparrow$ | FVD $\downarrow$ |
> | ----------- | ----------- | ----------- | ----------- | ----------- |
> | 1| first-frame | 0.477       | 16.628      | 1205.598    |
> | 2| first-frame + HOI | 0.555 | 18.930      | 864.739     |
> | 3| first-frame + text| 0.518 | 17.680      | 1063.458    |
> | 4| first-frame + text + HOI | **0.571** | **19.212** | **836.033**   |
>
> **A1.2 The quantitative comparison between X-Gen and ConsistI2V.**
>
> Note that, the design of our diffusion model follows SEINE[1], serving as an important baseline in our experiments. The improvement of X-Gen over SEINE is shown in Tables 1 and 5 in our manuscript:
> | Methods    | SSIM $\uparrow$ | PSNR $\uparrow$ | LPIPS $\downarrow$ | FVD $\downarrow$ |
> | ----------- | ----------- | ----------- | ----------- | ----------- |
> | SEINE | 0.518 | 17.680  | 0.321   | 1063.458    |
> | X-Gen | **0.537** | **18.395**  | **0.311**   | **1031.693**    |
>
> Regarding the comparison to ConsistI2V, despite a minor improvment in SSIM (0.532 vs. 0.537), X-Gen significantly outperforms ConsistI2V on perceptual metrics, i.e. LPIPS (0.351 vs. 0.311) and FVD (1109.314 vs. 1031.693).
>
> To further validate the effectiveness of our model, we replace the base architecture of the diffusion model with ConsistI2V and fine-tuned the base model for 10 epochs. Subsequently, an HOI mask encoder is trained based on the fine-tuned ConsistI2V model. During inference, we also feed the cross-view mask predictions into the generative model as the HOI condition.
>
> As shown in the Table below, introducing HOI to the ConsistI2V model consistently improves performance across all metrics. This demonstrates that **our approach generalizes well to different diffusion models, enhancing their ability to generate reliable HOI motions.**
>
> | Methods    | SSIM $\uparrow$ | PSNR $\uparrow$ | LPIPS $\downarrow$ | FVD $\downarrow$ |
> | ----------- | ----------- | ----------- | ----------- | ----------- |
> | ConsistI2V | 0.532 | 18.318  | 0.351   | 1109.314    |
> | ConsistI2V + X-Gen | **0.540** | **18.581**  | **0.314**   | **1017.377**  |
>
>
>
> ## **Q2. Adding the details on computational costs.**
>
> **A2.** We provide additional details on the training and inference costs of the cross-view mask prediction model (Seg) and video prediction model (Gen).
> Regarding the Gen model, we list the computational costs of the backbone (stage-1) and the mask encoder (stage-2).
> During training, we use 8 A100 GPUs with a batch size of 32. For inference, we employ a single A100 GPU and evaluate the model with a batch size of 1.
>
> | Methods    | Trainable params (MB) | Training time (sec/iter) | Inference speed (FPS) | Memory (GB) |
> | ----------- | ----------- | ----------- | ----------- | ----------- |
> | Seg model | 62 | 3.1  | 30   | 61    |
> | Gen model (stage-1) | 909 | 1.5  |1.1   | 66   |
> | Gen model (stage-2) | 277 | 1.3 | 0.9 | 52 |

---

> ### Author Response · Authors · 2024-11-23
> **(2/2) Response to Reviewer JxPc**
>
> ## **Q3. Performance on unpaired exo-ego videos and the usecase of paired data.**
>
> **A3.** This paper considers the translation from exocentric view to egocentric view, and thus the correspondence between exo-view and ego-view is required.
> This setting is useful in Embodied AI. For instance, a robot watches the the human demonstration of conducting an activity (e.g. cutting vegetables or washing dishes), it should map the exocentric demonstration to the egocentric view to learn and replicate the task in the same environment.
> The ability of AI assistants to provide visual instructions by matching third-person observations of fine-grained information from instructional videos to those in the user's first-person view is also underlined in Ego-Exo4D [2].
>
> To see how our approach performs in the case of unpaired ego-exo data,
> we evaluate the cross-view mask prediction model by choosing an unpaired exocentric video during inference, which leads to poor HOI segmentation performance as listed in the Table below. This highlights the alignment between different views.
> | Methods    | IoU $\uparrow$ | CA $\uparrow$ | LE $\downarrow$ |
> | ----------- | ----------- | ----------- | ----------- |
> | Aligned exo-ego | **20.3** | **0.207**  | **0.082** |
> | Unaligned exo-ego | 6.7 | 0.072  | 0.158 |
>
>
> [1]. Chen, Xinyuan, et al. "Seine: Short-to-long video diffusion model for generative transition and prediction." ICLR 2023.
>
> [2]. Grauman, Kristen, et al. "Ego-exo4d: Understanding skilled human activity from first-and third-person perspectives." CVPR 2024.

---

> ### Comment · Reviewer_JxPc · 2024-11-25
> **Rebuttal and Review Update**
>
> The rebuttal has addressed the questions I have. I will increase my rating.

---

### Meta-Review · Area_Chair_oJUp · 2024-12-19

**Metareview:**

The paper presents a framework for producing ego-centric videos of human-object interaction (HOI) conditioned on exo-centric videos, the first ego-centric frame, and a text description of the activity to be synthesized. This is a task that has many practical use cases, e.g., in a robotic imitation learning setting. The paper presents a two step approach for solving the task, where in the first step, HOI masks are generated auto-regressively using a cross-attention model between exo-ego video features and HOI mask features. These masks are then used in a diffusion model for video synthesis conditioned on the text. Experiments are provided on Ego-Exo4D and H2O benchmark and show promising results, including elaborate ablation studies analyzing the method on many architectural choices and aspects.

The paper received mainly positive reviews, with 2 borderline accepts and 1 accept. The reviewers overall liked the clarity in the organization and presentation in the paper, practical usefulness of the task, novelty in the presented architecture, and the elaborate experiments and ablation studies, including zero-shot generalization capabilities.

**Additional Comments On Reviewer Discussion:**

The reviewers also pointed out some important issues with the method and omissions in the experiments. The concerns are summarized as follows:
* *Reviewer JxPc* pointed out that the performance improvements reported in Table 1 are minor against a closely-related method ConsistI2V on SSIM and PSNR.
* *Reviewer 5j2P* pointed out missing ablation studies
* *Reviewer yjuR* requested additional details on test time compute and failure cases.
* All reviewers had concerns regarding the motivation for the specific architecture, how temporal dynamics is captured, and how HOI prediction is influencing the ego-video generation.

Authors provided a strong rebuttal addressing the above concerns to a reasonable level. Specifically,
* the authors pointed out that performances on LPIPS and FVD scores in Table 1 are higher as well as provided new results using ConsistI2V model as a baseline, demonstrating improvements thus responding convincingly to the issues pointed out by Reviewer JxPc.
* Authors also presented additional empirical results on the ablations requested by Reviewer 5j2P, clearly demonstrating benefits of the proposed methodology.
* Responding to Reviewer yjuR's concerns, authors detailed scenarios where the method may not work well, as well as details of compute and training/test time.

The reviewers were satisfied by the authors' responses and thereby raising their scores, inclining towards acceptance. While, the authors have addressed most of the concerns, AC accords with the sentiments of the reviewers that the motivation for the particular two-step approach, and in particular, the cross-attention model as depicted, are not sufficiently well-motivated, and better insights into why it leads to better prediction of the ego-frame from the exo-frame, may improve the quality of the work. Additional qualitative results could also have been presented that could improve the appreciation by the reviewers. That being said, the paper does address a useful task and the presented architecture seems to demonstrate promising results, with scope for further improvement, and thus AC recommends accept.

---

### Decision · Program_Chairs · 2025-01-22

Accept (Poster)